# Perioperative Management of a Pediatric Patient with Beckwith–Wiedemann Syndrome Undergoing a Partial Glossectomy According to Egyedi/Obwegeser

**DOI:** 10.3390/children10091467

**Published:** 2023-08-28

**Authors:** Antonio Izzi, Vincenzo Marchello, Aldo Manuali, Lazzaro Cassano, Andrea Di Francesco, Annalisa Mastromatteo, Andreaserena Recchia, Maria Pia Tonti, Grazia D’Onofrio, Alfredo Del Gaudio

**Affiliations:** 1UOC of Anesthesia and Resuscitation II, Fondazione IRCCS Casa Sollievo della Sofferenza, San Giovanni Rotondo, 71013 Foggia, Italy; antonioizzi1201@gmail.com (A.I.); vincenzomarchello@libero.it (V.M.); a.manuali@operapadrepio.it (A.M.); a.recchia@operapadrepio.it (A.R.); mariapiatonti@alice.it (M.P.T.); freddydelgaudio@libero.it (A.D.G.); 2UOC of Maxillofacial Surgery and Otolaryngology, Fondazione IRCCS Casa Sollievo della Sofferenza, San Giovanni Rotondo, 71013 Foggia, Italy; l.cassano@operapadrepio.it (L.C.); annalisamastromatteo@gmail.com (A.M.); 3UOS of Pediatric Maxillofacial Surgery, ASST Lariana, San Fermo della Battaglia, 22020 Como, Italy; andrea.difrancesco@asst-lariana.it; 4Health Department, Clinical Psychology Service, Fondazione IRCCS Casa Sollievo della Sofferenza, San Giovanni Rotondo, 71013 Foggia, Italy

**Keywords:** Beckwith–Wiedemann syndrome, perioperative management, obstructive sleep apnea syndrome

## Abstract

Here, we report the perioperative management of a clinical case of a 6 year, 5 month old girl suffering from Beckwith–Wiedemann syndrome undergoing a partial glossectomy procedure in a patient with surgical indication for obstructive sleep apnea syndrome (OSAS), difficulty swallowing, feeding, and speech. On surgery day, Clonidine (4 µg/kg) was administered. Following this, a general anesthesia induction was performed by administering Sevoflurane, Fentanyl, continuous intravenous Remifentanil, and lidocaine to the vocal cords, and a rhinotracheal intubation with a size 4.5 tube was carried out. Before starting the procedure, a block of the Lingual Nerve was performed with Levobupivacaine. Analgosedation was maintained with 3% Sevoflurane in air and oxygen (FiO_2_ of 40%) and Remifentanil in continuous intravenous infusion at a rate of 0.08–0.15 µg/kg/min. The surgical procedure lasted 2 h and 32 min. At the end of the surgery, the patient was under close observation during the first 72 h. In the pediatric patient with Beckwith–Wiedemann syndrome submitted to major maxillofacial surgery, the difficulty in managing the airways in the preoperative phase during intubation and in the post-operative phase during extubation should be considered.

## 1. Introduction

Beckwith–Wiedemann syndrome was discovered in 1964; the main clinical manifestations consist of overgrowth of the whole body or in part of it and an increased risk of developing childhood tumors in internal organs, such as the kidney and liver. Although much progress has been made in the diagnosis and treatment of the disease, there are still many aspects that remain to be understood.

Beckwith–Wiedemann syndrome is a rare genetic syndrome with an estimated prevalence of between 1/10,300 and 1/13,700 births, and an equal incidence in both sexes. In most cases (85% of cases), the transmission is unknown and, in the remaining percentage, an autosomal dominant transmission was found [1]. The Beckwith–Wiedemann spectrum is associated with molecular anomalies in a cluster of genes in the chromosomal region 11p15.5–11p15.4, subjected to genomic imprinting. In particular, in the areas IC1 and IC2, some genes with a specific activation/inactivation pattern that allows normal growth of the organism have been found. In 50% of cases, there is a hypomethylation of the maternal allele of IC2; in 5–10%, there is a hypermethylation of the paternal allele IC1. Other possible causes are represented by paternal uniparental isodisomy of the 11p15.5 region mutations in the CDKN1C gene and chromosomal abnormalities of 11p15. In about 20% of cases, no defect can be defined [2]. Normally, the diagnosis is made at birth; in other cases, it was possible to diagnose this pathology in the prenatal period or in the first years of life. It is a syndrome characterized by a particular phenotypic variability with possible visceral and somatic alterations with different expressiveness [3]. The possible visceral alterations that can be found can be cardiomyopathies, nephropathies (cortical or medullary cysts, hypercalciuria with renal lithiasis, vesico-ureteral reflux, recurrent urinary tract infections), hepatomegaly, splenomegaly, omphalocele, placental mesenchymal dysplasia, and condition of fetal adrenocortical cytomegaly, which is considered pathognomonic. Somatic anomalies consist of an overgrowth of certain areas of the body (macroglossia, increased ear folds, etc.), umbilical hernia or diastasis of the rectus abdominis muscles, facial naevus simplex, and a hemi-hypertrophy of an entire hemisome. Macroglossia is present in 95% of cases, with a recognizable facies of patients; a situation of neonatal hypoglycemia can also be observed in 30–50% (lasting about a week), as well as hyperinsulinism. In addition, subjects affected by this condition have a particular predisposition to the onset of malignant tumors in the first years of life (neuroblastoma, hepatoblastoma, rhabdomyosarcoma, unilateral Wilms tumor, adrenocortical carcinoma, or pheochromocytoma), a condition closely related to the molecular subtype [4]. It should be noted that cognitive development is usually normal in patients with Beckwith–Wiedemann syndrome. Since the first description in the 1960s, there have been several attempts to define specific criteria for clinical diagnosis. The international consensus has established that three different manifestations can be classified: the classical form, an isolated lateralized overgrowth, and an atypical form. The diagnosis of this pathology is made with the finding of at least three clinical signs among those listed above, but the confirmation is, in any case given, by molecular tests [5]. Prognosis is poor in the early stages of life, but tends to improve in patients who survive infancy. In fact, there are no strong data in the literature on relative indications for surgical correction of macroglossia in children affected by Beckwith–Wiedemann syndrome. Surgery should be considered only on the clinical status, especially if macroglossia is associated with other problems, such as tongue protrusion, airway obstruction, aesthetic appearance, persistent drooling, obstructive sleep apnea, feeding difficulties, problems with speech, and orthodontic and articulation problems such as malocclusion class III. In terms of the timing of surgery, if respiratory failures occur, surgery is suggested in neonatal period. In other situations, it is suggested before 2–5 years. When surgery occurs, the anesthetist must know this complex syndrome to avoid complications. The difficult management of the airways and the alteration of the metabolism must be considered during the anesthetic management. Airway obstruction in pediatric patients always represents a challenge and, particularly in patients with this syndrome, there is a potential aggravating factor for the possible presence of morphological alterations of the oral cavity with macroglossia [6].

## 2. Case Report

### 2.1. Patient Information

A week before surgery, the patient accompanied by her parents underwent a thorough anesthetic and surgical evaluation. A thorough medical history and physical examination are necessary to obtain useful information to avoid complications. A 6.5 year old girl, weight 24 kg, height 118 cm, was admitted to a tertiary hospital IRRCS Casa Sollievo della Sofferenza, San Giovanni Rotondo, Italy, for the surgical treatment of the maxillofacial malformation in Beckwith–Wiedmann syndrome. In her medical history, the child was born premature, but without a history of apnea, convulsions, or other pathological conditions. The diagnosis of Beckwith–Wiedemann syndrome was made at birth, when the patient was operated on for an omphalocele.

### 2.2. Clinical Findings

A month before the admission to the ward, a clinical and instrumental evaluation was planned by a multidisciplinary team formed by surgeons, anesthesiologists, and a psychologist. Neurologically, the patient presented as a normal infant of her age, a condition also confirmed by her parents, except for the difficulty in language and feeding. No pathological findings were found on cardiac and pulmonary auscultation, and these data were also confirmed by instrumental tests and diagnostics. The evaluation of the airways revealed a grade of Mallampati IV, with a mouth opening of 2 cm (Figure 1). The rest of the anthropometric parameters were not diriment. The blood oxygen saturation level (SpO_2_) was 99% in ambient air, the heartbeat was sinus, at a rate of 115 beats per minute, and the non-invasive blood pressure was 103/57 mmHg. The parents stressed the presence of significant nocturnal snoring when the child was sleeping. No abnormal laboratory findings, including coagulation, hepatic, and renal function indices were observed (Hb 14.4 g/dL, Plt 303.000/mL, WB 6.890, Cratinine 0.5 mg/dL, Glycemia 85 mg/dL, Azotemia 25 mg/dL, Bilirubin 0.9 mg/dL, GOT 32 U/L, GPT 38 U/L, PT 107%, INR 0.97 ratio, PTT 27.8 sec, Protein 7.9 g/dL).

### 2.3. Timeline and Therapeutic Intervention

All the clinicians had experience in treating severe adult and pediatric maxillofacial dimorphisms. Surgeons proposed the technique to the parents, and explained the complications of the maxillofacial surgery necessary to correct their child’s facial malformation. The expert anesthesiologists proposed general anesthesia, and explained potential risks of anesthesiology management according to the difficult airway management and metabolic implications of the syndrome. Also, the need for post-operative admission to Intensive Care for appropriate treatment was explained. The psychologist also took care of the whole family with their consent from the beginning of the hospital stay. The day before the surgery, the multidisciplinary team had a briefing to discuss the intraoperative management strategy. The difficult airway management was an important aspect for the expected difficulty to ventilate and to intubate, especially for the encumbrance of the lingual body (Figure 2). The expert anesthesiologist and the anesthesia nurses assessed the pediatric difficult airway algorithms and checked the difficult intubation trolley (face masks of different sizes, nasal cannulas, pediatric orotracheal cannulas, pediatric bronchoscope, laryngeal masks of different sizes, and emergency tracheostomy sets) available in operating room. On the day of the surgery, the patient, with her mother, after a negative nasal swab for SARS-CoV-2, was admitted to the maxillofacial surgery ward. The patient was re-evaluated by the anesthetist and the surgeon, and her vital parameters (SpO_2_, T°, NIBP and 3-lead ECG) were re-measured; all were within normal limits. Clonidine (4 µg/kg) was administered as premedication intranasally half an hour before the surgery through a nebulizer (MAD), which also helps with an antiscialogogue effect, under constant monitoring and two drops per nostril of a nasal vasoconstrictor to avoid potential bleeding during tube passage. She was accompanied to the operating block together with the parent and the psychologist, in order to reassure her. After the sign-in and timeout assessment and compilation of the Surgical Safety checklist, the general anesthesia induction was performed with an inhalatory technique by administering Sevoflurane in increasing doses up to a concentration of 8%. When loss of consciousness occurred, the mother was accompanied outside the operating block, and two peripheral venous accesses were inserted. Fentanyl (1 µg/kg ev) was given, and continuous intravenous Remifentanil administration started (0.05 µg/kg/min). The patient was pre-oxygenated with nasal cannulae and, as soon as she stopped breathing, she was manually ventilated, minimizing the period of hypo-oxygenation. In addition to the previous monitoring, electrodes were placed for the evaluation of entropy maintained until the end of the intervention, as well as the monitoring for the end tidal CO_2_ with visualization of the capnographic curve and the temperature probe. Devices were used to warm the patient (a mattress with a heater and an infusion of hot liquids) in the recovery room (pre-warming) and throughout the surgery. Laryngoscopy with videolaryngoscope blade 2 was performed, with a match of grade II to the Cormack-Lehane classification; 40 mg of 2% lidocaine was administered to the vocal cords. After a few minutes, during which the patient continued to be manually ventilated with 100% O_2_, and after analgosedation, rhinotracheal intubation with a size 4.5 tube was carried out. The procedure was successful on the first attempt, and did not lead to problems of any kind. The patient maintained stable vital signs and, in particular, an SpO_2_ of no less than 98% throughout the procedure. The tube was then fixed to the nasal choana through a point, to avoid accidental dislocations by the operators, given the area proximal to the operating field. Anti-decubitus devices (Liofoam) were placed on the skin where there could be contact with the tube. A bladder catheter and a gastric nose tube were placed, and a radial artery was cannulated with the Seldinger technique for invasive blood pressure monitoring. Finally, before starting the procedure, a block of the Lingual Nerve was performed with Levobupivacaine 0.5% 1 mL bilaterally. Analgosedation was maintained with 3% Sevoflurane in air and oxygen (FiO_2_ of 40%), and Remifentanil in continuous intravenous infusion at a rate of 0.08–0.15 µg/kg/min. The vital parameters continued to remain stable throughout the procedure, blood losses were contained thanks above all to the experience of the operator, and the water balance was constantly monitored. An initial blood gas analysis was performed, then one after the first hour of surgery and one before leaving the operating block, and all three did not show any noteworthy changes. The procedure lasted 2 h and 32 min; the type of intervention performed consisted of a partial glossectomy according to the Egyedi/Obwegeser technique, in which the central portion of the lingual body is resected through a hourglass cut and the side flaps are sutured (Figure 3 and Figure 4). The result will certainly be a reduction in the volume of the organ, with a slightly pointed final shape and a significant limitation of the encumbrance of the tongue within the oral cavity. At the end of the surgery, the patient was transferred to intensive care, analgosedated, and mechanically ventilated with a portable ventilator, always under monitoring of vital parameters.

### 2.4. Follow-Up and Outcomes

In the first 72 h of hospitalization. the patient was under close observation with her head positioned at 30° in order to reduce edema; during the first 48 h, the patient was kept asleep and with protected airways by continuous infusion of Propofol (with a TCI model Paedifusor to the effect of 2–3 µg/mL) and Remifentanil (0.05–0.1 µg/kg/h).

The re-evaluation of the surgical wound and blood loss was performed daily by the surgeons, in addition to the control of blood chemistry and clinical parameters by the resuscitators. At the end of the second day, Propofol was suspended, and the dosage of Remifentanil was reduced until consciousness and spontaneous breathing were restored. The patient was extubated in a protected manner and administered oxygen with high flow nasal cannulae at 50 L/min FiO_2_ of 50%, at 31 °C. After extubation, her respiratory and hemodynamic conditions were stable. Pain was assessed by a behavioral scale showing a mild discomfort, but no pain. During the first day after extubation, Dexmetomidine was administered in low doses (maximum 0.7 µg/kg/h) together with Remifentanil (0.05 µg/kg/min), to guarantee the patient’s tolerance of the high flows and the nasogastric tube, which remained in place to allow early enteral refeeding. At day 1, the child was awake, conscious, and with stable vital parameters. At day 4, she was able to do daily activities such as watching television and playing with a tablet for a complete neurocognitive recovery. On the fifth day, she was transferred to the pediatric ward and discharged from there the following day, with the indication to begin rehabilitation with a speech therapist as soon as possible and to carry out pedodontic checks over time.

## 3. Discussion

According to our opinion, the perioperative management of this clinical case is interesting for the possible consideration of the severe complications which could occur during and after major surgery in patients with Beckwith–Wiedemann syndrome. The multidisciplinary approach is mandatory for the management of this complex surgery. In the preoperative phase, a complete assessment of the patient’s clinical and morphological conditions is essential to collect a complete medical history. Also, the psychological aspects of a demolitive, painful facial surgery limiting feeding and specking are important. The work carried out by the psychologist present during the interviews with family members is to reassure the parents and, indirectly, the child. On the day of surgery, a quick re-evaluation avoids encountering adverse conditions during the anesthesia induction phase (e.g., onset of infections, colds, increased secretions, etc.). There is little literature on the subject of Premedication with off-label Clonidine in pediatric patients; in our service, this drug is used routinely both in premedication and in the context of intensive care. In recent years, its use has increased, which allows anxiolysis and prevents drooling, essential in the context of a patient with these problems, and helps the analgesic action of intra- and post-operative opioids [7]. In the case of a patient in which a reduced oxygen reserve is observed, as in this case, it is possible to take advantage of the pharmacodynamics of Clonidine, which allows one to avoid the depression of the respiratory drive, as opposed to other sedative drugs such as Midazolam. The use of a nasal vasoconstrictor avoids the onset of a dangerous event, such as bleeding during the passage of the tube in the nose, causing a reduction in vision and the tarnishing of the optics of the video laryngoscope used in this case, in a patient who already has a considerable bulk in the mouth [8]. Pre-oxygenation with nasal cannulae with 100% O_2_ minimizes the risk of desaturation during the intubation procedure. The fibrobronchoscopy, in case of an inability to perform a valid laryngoscopy or a difficulty in ventilation and/or intubation, was present and available in the operating room, as was a cricotomy kit [9]. The use of these devices, which necessarily involves adequate training, is an essential skill in all areas of difficult airway management. The administration of high concentrations of Sevoflurane or muscle relaxants to patients in the supine position could cause the tongue to fall into the retro-lingual space, with severe airway obstruction during induction of anesthesia. For these reasons, ventilation was performed using a face mask using Sevoflurane, gradually increasing its concentration until an adequate depth of anesthesia was obtained. Curaries are not considered routine drugs in the intubation of pediatric patients; in this context, the use of a local anesthetic sprayed directly on the vocal cords through a special cannula was preferred [10]. Lingual Nerve Block is an effective and easy-to-implement analgesic technique [11], and has potentially reduced intra- and post-operative opioid dosages according to a policy of opioid spearing and preemptive analgesia [12]. Complete intraoperative monitoring seemed necessary to minimize the risks, in particular the constant evaluation of the temperature, a particularly important parameter in a pediatric patient, given the ease of heat loss [13], and a control of blood loss, given the very perfused surgical area. Regarding the surgical technique, there are various options that can be used in this context. The goal above all is to achieve an almost normal level of lingual anatomy both in terms of size and shape, such that the organ remains behind the incisors with sufficient mobility to moisturize the lips [14]. In our case, the technique chosen was a partial glossectomy according to Egyedi and Obwegeser, which consists of a symmetrical resection (see case report), and is particularly suitable for bilateral macroglossia. According to the experience of the operators, this method guarantees a better functionality of the tongue in the long term, both for language and for food; it also reduces the risk of intraoperative bleeding as well as avoiding the collapse of the tissues of the soft palate and tongue, reducing the risk of post-operative airway obstruction [15]. Balaji et al. claim that “the main stay of surgical treatment of macroglossia is to provide a tongue that can function in the most efficient aspect in terms of form and function”, and they proposed a classification of efficient management of macroglossia; for situations where the length and width abnormalities occur, Egyedi–Obwegeser offers a solution [14]. Recently, in 2023, The American Association of Oral and Maxillofacial Surgeons presented an algorithm for the management of idiopathic macroglossia. The Egyedi and Obwegeser technique is described as a technique used in many patients, because it allows both narrowing and shortening of the tongue and preserves the lingual nerve and hypoglossal nerves, whereas the loss of the tip has not been reported to result in a marked sensory deficiency [16]. Consequently, in more recent studies, we described the Egyedi and Obwegeser technique as safer for tongue reduction in our little patient. Macroglossia is definitely a growth stimulus for the jaw; in fact, macroglossia could be a factor of structural and functional changes in the growth of the facial mass and, in particular, of the mandible, to compensate for the largeness of the tongue. However, whether this is the only trigger for prognathism, or if the genetic component contributes an additional role, is not known [17]. It can be considered that an early intervention to reduce the tongue is recommended to reduce the risk of structural alteration of the stomatognathic system and to develop both the mandibular prognathism and the anterior open bite, in addition to any social and psychological problems for the child. The optimal age for surgery is unclear; in general, it is recommended to perform the reduction of the tongue after one year of age to reduce the risk of perioperative complications, if there is no condition that involves problems for the child’s survival [18]. No noticeable changes and/or complications were observed in the post-operative phase. Transportation to the ICU was mandatory due to the risk of tongue edema and bleeding. According to our experience, the peak of lingual edema is observed within the first 48 h. Daily surgical re-evaluation is essential to avoid these complications. After extubation, the positioning of high flow nasal cannula (HFNC) in this patient reduces the risk of respiratory crises, as described in the literature [19]. An HFNC application decreases nasopharyngeal dead space and increases the alveolar fraction of oxygen, reduces work of breathing and respiratory muscle fatigue, improves management of respiratory secretions, and reduces upper airway obstruction episodes through the humidified oxygen and the positive end expiratory pressure applied to the airway [20,21]. An alpha 2 agonist, such as Dexmetomidine, and an opioid, in this case Remifentanil, have receptor synergism, as reducing the dosages will achieve similar or higher efficacy than drugs administered individually, and will have fewer side effects (Figure 5) [22]. Furthermore, by exploiting the pharmacodynamics of Dexmetomidine, we obtain a valid sedation while maintaining spontaneous breathing. As regards Remifentanil being an insensitive context drug with a very rapid off-set, we have the possibility of modulating analgesia and reducing or increasing the dosage in a continuous infusion without the problem of accumulation or excessive respiratory depression [23]. Paracetamol was imbricated following extubation, in the control of post-operative analgesia, by administering 350 mg intravenously up to three times a day, observing a valid result. Rapid enteral refeeding, early mobilization, and discharge in the shortest possible time in accordance with the indications of the ERAS protocol indicate a reduction in long-term complications and the onset of infections in the patient in intensive care or in any case following surgery [24]. Rehabilitation followed by consultation with a speech therapist is essential for the child to resume normal daily life for sociability, language, and cognitive development in an age of growth in which these aspects are very important. Finally, a conservative functional orthodontic treatment could be useful to prevent dental and skeletal malformations and lead to a regular craniofacial physiognomy over the years [25].

## 4. Conclusions

In a pediatric patient with Beckwith–Wiedemann syndrome submitted to a major maxillofacial surgery, a multidisciplinary approach is needed to evaluate anatomic, metabolic, and psychological aspects. The difficult airway management, the potential alteration of metabolic pathways, the surgical technique, and the psychological and rehabilitative approaches can lead to severe complications if underestimated. Therefore, given the complexity of the intervention and the non-routine nature of this, at least in our hospital, it is considered mandatory to take all the necessary precautions to deal with any emergency situations, as well as to implement prophylaxis techniques to avoid such situations.

## Figures and Tables

**Figure 1 children-10-01467-f001:**
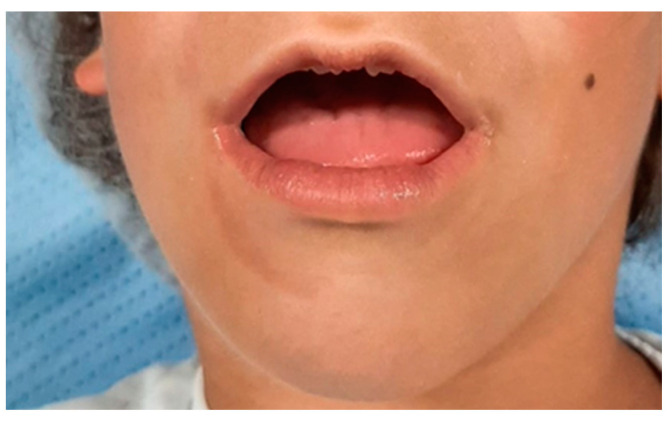
Interdental distance preoperative assessment.

**Figure 2 children-10-01467-f002:**
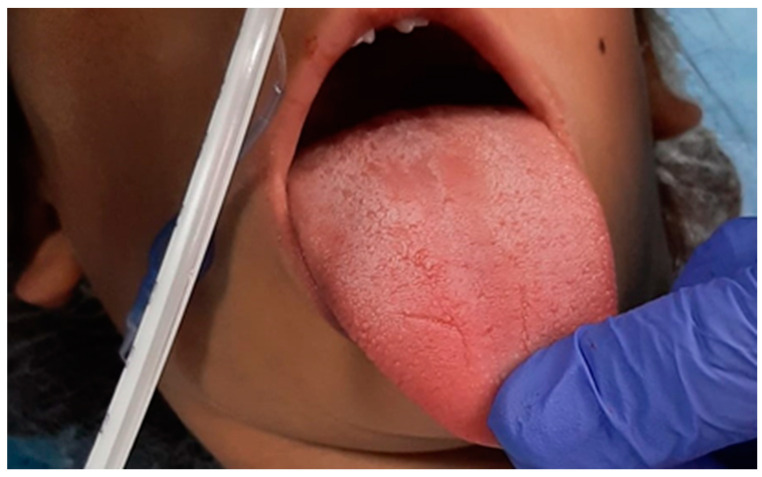
Tongue size at preoperative assessment.

**Figure 3 children-10-01467-f003:**
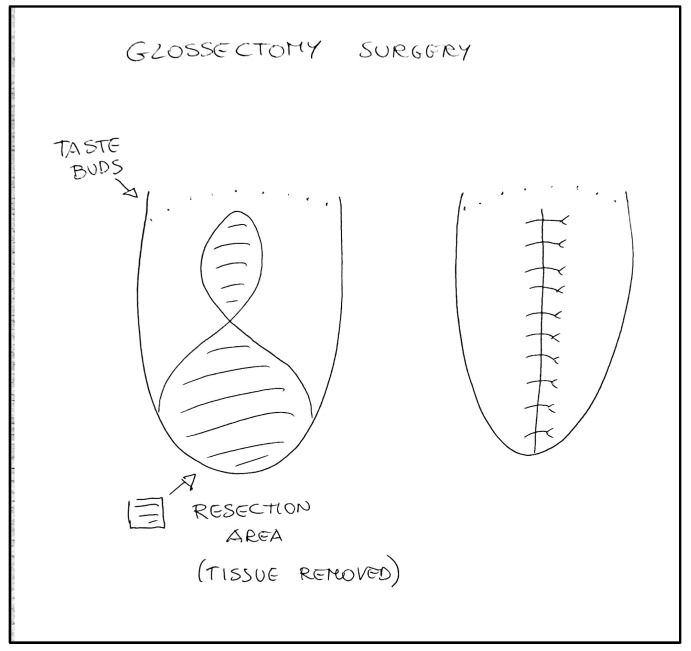
Drawing performed for preoperative sharing of the type of surgery.

**Figure 4 children-10-01467-f004:**
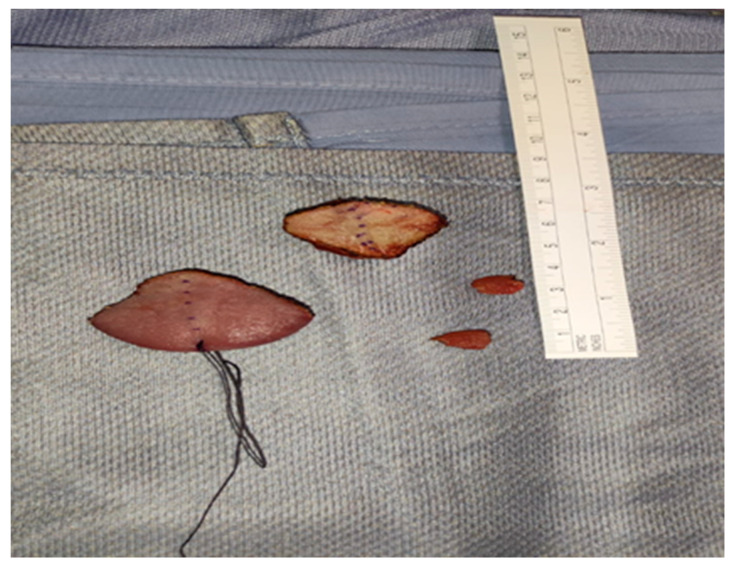
Surgical parts of the tongue removed.

**Figure 5 children-10-01467-f005:**
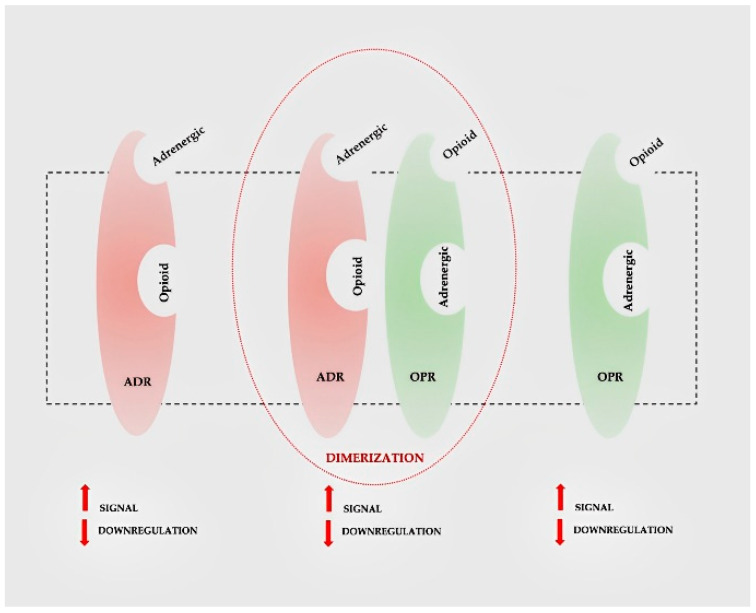
Opioid and adrenergic receptors.

## Data Availability

Not applicable.

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
