# Peer review of "Perioperative Management of a Pediatric Patient with Beckwith–Wiedemann Syndrome Undergoing a Partial Glossectomy According to Egyedi/Obwegeser"

_children, 2023, doi:10.3390/children10091467_

Round 1

Reviewer 1 Report

The authors present a case report on the perioperative management of a clinical case of a child with Beckwith-Wiedemann syndrome undergoing a partial glossectomy. Although this case is interesting, several major concerns need to be addressed.

1. The abstract needs to be rewritten. Include a short introduction of the problem the author wants to address and a brief conclusion.

2. The case report section is too lengthy. Please stick to the important part. How was the syndrome diagnosed? Was PSG carried out? Any role of overnight pulse oximetry?

3. How was the lingual nerve block given?

4. Include surgical steps with pictures. How was the incision closed?

5. Include a post-operative picture.

6. How was the patient during the follow-up?

7. Include literature review on other surgical techniques and why this technique is superior?

Extensive grammatical mistakes are encountered.

Author Response

The authors present a case report on the perioperative management of a clinical case of a child with Beckwith-Wiedemann syndrome undergoing a partial glossectomy. Although this case is interesting, several major concerns need to be addressed.

We thank the Reviewer for his/her thoughtful and constructive comments. We have addressed each of the issues raised and have highlighted the relevant revisions in the manuscript itself. Below, please find item-by-item responses to the Reviewer’s comments, which are included verbatim.

  1. The abstract needs to be rewritten. Include a short introduction of the problem the author wants to address and a brief conclusion.                      1. According to the reviewer, we rewrote the Abstract section.

  1. The case report section is too lengthy. Please stick to the important part. How was the syndrome diagnosed? Was PSG carried out? Any role of overnight pulse oximetry?                                                                                2. According to reviewer observation, we divided the Case Report section in subsection in order to make the clinical case presented more readable and clear. About diagnosis of the syndrome, it was carried out previously through genetic tests and thanks to the evolution of the patient's growth of symptoms and diriment signs. In particular from the point of view of problems concerning the airways the PSG had not been performed and nocturnal pulse oximetry was not monitored, but the diagnosis of OSAS was simply made thanks to the observation of the symptoms by the parents during the night and by the child herself who complained of sleep disturbances.

  1. How was the lingual nerve block given?                                                       3. After moving the tongue medially, the needle is inserted under the mucosa at the base of the pillar, 0.5 cm lateral to the base of the tongue bilaterally; by injecting 1ml of 0.5% Levobupivacaine per side.

  1. Include surgical steps with pictures. How was the incision closed?              4. Unfortunately we have no pictures about surgical steps. The resection was performed by removing the anterior medial portion of the lingual body and the tip of the tongue, trying to spare as much as possible the innervation and gustatory sensitivity. The reconstruction took place by rejoining the two lateral ends and repackaging a lingual structure as aesthetically and functionally valid as possible, considering the possible and inevitable losses necessary due to the nature of the procedure itself.

  1. Include a post-operative picture.                                                                  5. Unfortunately we have no picture about post-operative time.

  1. How was the patient during the follow-up?                                                   6. One year after the patient's discharge, a marked improvement in her quality of life was noted. There was an improvement in speech, in interaction with his peers and with his parents, and there was also a reduction in swallowing, sleeping and breathing problems.

  1. Include literature review on other surgical techniques and why this technique is superior?                                                                                   7. Regarding the surgical technique, there are various options that can be used in this context. The goal of all is to achieve an almost normal level of lingual anatomy both in terms of size and shape, such that the organ remains behind the incisors with sufficient mobility to moisturize the lips [15]. In our case the technique chosen was a partial glossectomy according to Egyedi and Obwegeser which consists of a symmetrical resection (see case report) and is particularly suitable for bilateral macroglossia. According to the experience of the operators, this method guarantees a better functionality of the tongue in the long term, both for language and for food, it also reduces the risk of intraoperative bleeding as well as avoiding the collapse of the tissues of the soft palate and tongue, reducing the risk of post-operative airway obstruction [16]. Balaji et al claims that “the main stay of surgical treatment of macroglossia is to provide a tongue that can function in the most efficient aspect in terms of form and function” and he proposed a classification of efficient management of macroglossia: for situations where the length and width abnormalities occur, Egyedi-Obwegeser offers a solution [17]. Recently, in 2023, The American Association of Oral and maxilofaccial Surgeons presented an algorithm for the management of idiopathic macroglossia. Egyedi and Obwegeser technique is described as a technique used in many patients because it allows both narrowing and shortening of the tongue and preserves the lingual nerve and hypoglossal nerves whereas the loss of the tip has not been reported to result in a marked sensory deficiency [18]. Consequently on more recent studies, we describe the Egyedi and Obwegeser technique as the safer for the tongue reduction in our little patient.                                                                                                                       17. Balaji SM. Reduction glossectomy for large tongues. Ann Maxillofac Surg. 2013;3(2):167-172. doi:10.4103/2231-0746.119230: Based on a single center experience, I propose a classifi cation of efficient management of macroglossia. In cases where length of the tongue is abnormal, resection designs of Reinwald or Gupta or Harris, Blair, Hendrick or Dingman and Grabb can be employed. In situations where abnormal width is a primary concern, Edgerton/Butlin and Handley/Pichler design or their modifi cation can be employed. For isolated thickness abnormalities, horizontal resections and key-hole debulking procedures are recommended. For situations where the length and width abnormalities occur, Butlin and Ensign or Magee or Kole or Egyedi-Obwegeser or Deplange or Pless or Austermann or Machtens designs and their modifi cations offer solution. For abnormally wide and thick tongue, dorsal resections are the only choice. In rare situations, when the length, width, and thickness are abnormal, Heggie’s Stellate anterior wedge or Davalbhakta and Lamberty design offer best possible solution. It has been already documented that there is no single ideal tongue resection procedure, rather the procedure needs to be customized considering the etiology, age, gender, existing dimension, and postoperative form/dimension desired. Such individual approach gives more pleasing results than the predetermined ones. In case of older individuals with mild to moderate enlargement, peripheral surgical trimming is ideal to prevent loss of taste as well as speech abnormalities. In case of extreme enlargement, AWR and CR can be chosen but with caution. The underlying disorder has to be corrected.

  1. Extensive grammatical mistakes are encountered.                                           8. According to reviewer suggestion, we re-obtained a review of the manuscript from a native English speaker.

Reviewer 2 Report

Thank you for inviting me to review this manuscript on hot topic Perioperative management of a pediatric patient with Beckwith-Wiedemann Syndrome to undergo partial glossectomy according to Egyedi/Obwegeser. Beckwith-Wiedemann syndrome (BWS) is a growth disorder variably characterized by macroglossia, hemihyperplasia, omphalocele, neonatal hypoglycemia, macrosomia, embryonal tumors. The manuscript is easy to read and all data and conclusions presenting huge interest with impact on clinical practice.

I have several  proposals:

1.     Please substitute Figure 3. Drawing performed for preoperative sharing of the type of surgery with picture in Photoshop or any other program

2.      Conclusions can be more shorter and more exact, now its very long

Author Response

Thank you for inviting me to review this manuscript on hot topic Perioperative management of a pediatric patient with Beckwith-Wiedemann Syndrome to undergo partial glossectomy according to Egyedi/Obwegeser. Beckwith-Wiedemann syndrome (BWS) is a growth disorder variably characterized by macroglossia, hemihyperplasia, omphalocele, neonatal hypoglycemia, macrosomia, embryonal tumors. The manuscript is easy to read and all data and conclusions presenting huge interest with impact on clinical practice.

I have several  proposals.

We thank the Reviewer for his/her thoughtful and constructive comments. We have addressed each of the issues raised and have highlighted the relevant revisions in the manuscript itself. Below, please find item-by-item responses to the Reviewer’s comments, which are included verbatim.

  1. Please substitute Figure 3. Drawing performed for preoperative sharing of the type of surgery with picture in Photoshop or any other program.         1. We re-drawn Figure 3.

  1. Conclusions can be more shorter and more exact, now it’s very long.         2. The Conclusion section has been reduced.

Reviewer 3 Report

The text you provided is a comprehensive review of the perioperative management in a pediatric patient with Beckwith-Wiedemann syndrome undergoing major maxillofacial surgery. It covers various aspects of the case, including medical history, preoperative evaluation, anesthesia induction, surgical technique, and postoperative care. The multidisciplinary approach, attention to detail, and the use of appropriate anesthesia and pain management techniques are well-presented and appear to align with the objectives of the review.

Overall, the text achieves its intended objectives and effectively describes the management of the specific case. It provides valuable insights into the challenges and considerations involved in managing a patient with Beckwith-Wiedemann syndrome during surgery. However, it could benefit from some improvements, such as a more critical discussion of potential complications and a broader literature review on similar cases or alternative surgical techniques. These additions could enhance the overall depth and impact of the review.

As with any piece of writing, there is always room for further improvement and refinement. Depending on the intended audience and publication platform, you may consider peer review or further editing to strengthen the text's clarity, coherence, and overall impact.

Author Response

The text you provided is a comprehensive review of the perioperative management in a pediatric patient with Beckwith-Wiedemann syndrome undergoing major maxillofacial surgery. It covers various aspects of the case, including medical history, preoperative evaluation, anesthesia induction, surgical technique, and postoperative care. The multidisciplinary approach, attention to detail, and the use of appropriate anesthesia and pain management techniques are well-presented and appear to align with the objectives of the review.

We thank the Reviewer for his/her thoughtful and constructive comments. We have addressed each of the issues raised and have highlighted the relevant revisions in the manuscript itself. Below, please find item-by-item responses to the Reviewer’s comments, which are included verbatim.

  1. Overall, the text achieves its intended objectives and effectively describes the management of the specific case. It provides valuable insights into the challenges and considerations involved in managing a patient with Beckwith-Wiedemann syndrome during surgery. However, it could benefit from some improvements, such as a more critical discussion of potential complications and a broader literature review on similar cases or alternative surgical techniques. These additions could enhance the overall depth and impact of the review. As with any piece of writing, there is always room for further improvement and refinement. Depending on the intended audience and publication platform, you may consider peer review or further editing to strengthen the text's clarity, coherence, and overall impact.
  1. We improved the Discussion section by a more critical comparison with other surgical techniques, and re-wrote the Abstract section, according to reviewer suggestion.

Reviewer 4 Report

Good day,

Dear Authors,

Thank You for opportunity to read Your manuscript.

I have several offers and comments to improve it.

Firstly, it is not very clear why You attached not filled agreement for anesthesia especially on Italian.

Abstract

Please, check the requirements for abstract in ‘Children’. It has not enough words. Also, there is no logical conclusion.

For description of clinical case, it is important to follow requirements according to CARE (https://www.equator-network.org/reporting-guidelines/care/). Please, change the article structure according to CARE and attach the finished check list in non-published materials section.

Figure 3 is too much ‘handmade’. Please, put here readable and clear scheme or several intraoperative photos with these steps.

Please, re-write figure 5 legend similar with other figures.

There is no need in table 1, its content must be put in appropriate place in the article text.

Discussion

One case is not represented in the complications avoiding. Please, re-write this sentence.

Also, it is not clear why for one girl treatment there is need in 10 people.. The contributors roles are discussable.

Sincerely, Reviewer

Author Response

Good day,

Dear Authors,

Thank You for opportunity to read Your manuscript.

I have several offers and comments to improve it.

We thank the Reviewer for his/her thoughtful and constructive comments. We have addressed each of the issues raised and have highlighted the relevant revisions in the manuscript itself. Below, please find item-by-item responses to the Reviewer’s comments, which are included verbatim.

  1. Firstly, it is not very clear why You attached not filled agreement for anesthesia especially on Italian.                                                                    1. On July 21, 2023 the Assistant Editor was asked to us, via email, the written inform consent file in order to further process our manuscript.

  1. Abstract

Please, check the requirements for abstract in ‘Children’. It has not enough words. Also, there is no logical conclusion.

  1. According to reviewer observation, we re-wrote the Abstract section.

  1. For description of clinical case, it is important to follow requirements according to CARE (https://www.equator-network.org/reporting-guidelines/care/). Please, change the article structure according to CARE and attach the finished check list in non-published materials section.         3. By Reviewer suggestion, we structured the article according to CARE guidelines.

  1. Figure 3 is too much ‘handmade’. Please, put here readable and clear scheme or several intraoperative photos with these steps.                          4. We re-drawn Figure 3.

  1. Please, re-write figure 5 legend similar with other figures.                           5. We re-wrote the Figure 5 legend, according to reviewer suggestion.

  1. There is no need in table 1, its content must be put in appropriate place in the article text.                                                                                               6. We removed Table 1 and listed the difficult intubation pediatric trolley in the text, as suggested by the reviewer.

  1. Discussion

One case is not represented in the complications avoiding. Please, re-write this sentence.

  1. According to reviewer observation, we re-wrote the sentence as shown below: “According to our opinion the perioperative management of this clinical case is in-teresting for avoiding possible consideration of the severe complications which could occur during and after major surgery in patients with Beckwith-Wiedemann Syndrome”.

  1. Also, it is not clear why for one girl treatment there is need in 10 people. The contributors roles are discussable.                                                         8. The difficulty in managing the case also emerges from the need to use more resources. The importance of a multidisciplinary approach was underlined in the case report in order to evaluate anatomic, metabolic and psychological aspects, and manage the patient and her family before, during and after surgery.

Round 2

Reviewer 1 Report

The authors have adequately revised the manuscript. I would suggest the inclusion of a table with a literature search on outcomes in children operated with similar techniques. The outcome can be measured in complications and in terms of airway, swallowing and feeding.

Author Response

Thank you for your valuable suggestion. We proceeded to carry out research on this after his advice and as already observed in writing this case report, there are few cases and evaluations of this type of intervention recorded in the literature other than those already mentioned in this article; in particular we have found that the percentage of intra and post-operative problems are minor and above all due to a condition of non-management of the airways or bleeding problems, probably due to lack of experience in this regard, or in conditions in which the centers involved have little experience with pediatric ICU management.

Reviewer 3 Report

The article discusses the perioperative management of a pediatric patient with Beckwith-Wiedemann syndrome undergoing major maxillofacial surgery. The syndrome is characterized by macroglossia (enlarged tongue) among other physical and genetic anomalies. The article presents a detailed case report of a 6.5-year-old girl with this syndrome, outlining the clinical findings, preoperative evaluations, anesthesia management, surgical technique, postoperative care, and outcomes.

Strengths:

1. Comprehensive Case Presentation: The article provides a detailed account of the patient's medical history, clinical evaluation, surgical approach, anesthesia management, and postoperative care. This comprehensive information is beneficial for medical professionals looking to understand the challenges and considerations in managing similar cases.

2. Multidisciplinary Approach: The article highlights the importance of a multidisciplinary approach involving surgeons, anesthesiologists, psychologists, and other specialists. This approach is crucial for addressing the diverse challenges posed by the patient's medical condition and surgery.

3.  Technical Details: The article delves into technical aspects such as medication choices, induction techniques, airway management, and analgesia strategies. These details can be valuable for healthcare professionals dealing with complex cases.

Weaknesses:

Lack of General Context: The article assumes a certain level of familiarity with Beckwith-Wiedemann syndrome, its prevalence, and its clinical manifestations. A brief introduction or background section could provide context for readers less familiar with the syndrome.

Lack of Comparative Analysis: The article presents a single case report without comparing the results to other cases or discussing potential variations in management approaches. A broader discussion of similar cases or alternative management strategies could enhance the article's relevance.

Overall, the article provides an insightful glimpse into the perioperative management of a pediatric patient with Beckwith-Wiedemann syndrome undergoing maxillofacial surgery. While the case presentation is detailed and informative, the article could benefit from providing more general context, simplifying technical language, addressing ethical considerations, and incorporating references to support the discussed practices.

Author Response

The article discusses the perioperative management of a pediatric patient with Beckwith-Wiedemann syndrome undergoing major maxillofacial surgery. The syndrome is characterized by macroglossia (enlarged tongue) among other physical and genetic anomalies. The article presents a detailed case report of a 6.5-year-old girl with this syndrome, outlining the clinical findings, preoperative evaluations, anesthesia management, surgical technique, postoperative care, and outcomes.

Strengths:

  1. Comprehensive Case Presentation: The article provides a detailed account of the patient's medical history, clinical evaluation, surgical approach, anesthesia management, and postoperative care. This comprehensive information is beneficial for medical professionals looking to understand the challenges and considerations in managing similar cases.
  2. Multidisciplinary Approach: The article highlights the importance of a multidisciplinary approach involving surgeons, anesthesiologists, psychologists, and other specialists. This approach is crucial for addressing the diverse challenges posed by the patient's medical condition and surgery.
  3. Technical Details: The article delves into technical aspects such as medication choices, induction techniques, airway management, and analgesia strategies. These details can be valuable for healthcare professionals dealing with complex cases.

We thank the Reviewer for his/her thoughtful and constructive comments. We have addressed each of the issues raised and have highlighted the relevant revisions in the manuscript itself. Below, please find item-by-item responses to the Reviewer’s comments, which are included verbatim.

Weaknesses:

1. Lack of General Context: The article assumes a certain level of familiarity with Beckwith-Wiedemann syndrome, its prevalence, and its clinical manifestations. A brief introduction or background section could provide context for readers less familiar with the syndrome.                                                                                   1. According to reviewer suggestion, we added a brief introduction about Beckwith-Wiedemann syndrome.

  1. Lack of Comparative Analysis: The article presents a single case report without comparing the results to other cases or discussing potential variations in management approaches. A broader discussion of similar cases or alternative management strategies could enhance the article's relevance.                                                                                                      2. About management strategy, there are various options that can be used in this context. We included the following paragraph in Discussion section:

-The goal of all is to achieve an almost normal level of lingual anatomy both in terms of size and shape, such that the organ remains behind the incisors with sufficient mobility to moisturize the lips [15]. In our case the technique chosen was a partial                                               glossectomy according to Egyedi and Obwegeser which consists of a symmetrical resection (see case report) and is particularly suitable for bilateral macroglossia.

According to the experience of the operators, this method guarantees a better functionality of the tongue in the long term, both for language and for food, it also reduces the risk of intraoperative bleeding as well as avoiding the collapse of the tissues of the soft palate and tongue, reducing the risk of post-operative airway obstruction [16].

Balaji et al claims that “the main stay of surgical treatment of macroglossia is to provide a tongue that can function in the most efficient aspect in terms of form and function” and he proposed a classification of efficient management of macroglossia: for situations where the length and width abnormalities occur, Egyedi-Obwegeser offers a solution [17]. Recently, in 2023, The American Association of Oral and maxilofaccial Surgeons presented an algorithm for the management of idiopathic macroglossia. Egyedi and Obwegeser technique is described as a technique used in many patients because it allows both narrowing and shortening of the tongue and preserves the lingual nerve and hypoglossal nerves whereas the loss of the tip has not been reported to result in a marked sensory deficiency [18]. Consequently on more recent studies, we describe the Egyedi and Obwegeser technique as the safer for the tongue reduction in our little patient.-

  1. Balaji SM. Reduction glossectomy for large tongues. Ann Maxillofac Surg. 2013;3(2):167-172. doi:10.4103/2231-0746.119230: Based on a single center experience, I propose a classification of efficient management of macroglossia. In cases where length of the tongue is abnormal, resection designs of Reinwald or Gupta or Harris, Blair, Hendrick or Dingman and Grabb can be employed. In situations where abnormal width is a primary concern, Edgerton/Butlin and Handley/Pichler design or their modifi cation can be employed. For isolated thickness abnormalities, horizontal resections and key-hole debulking procedures are recommended. For situations where the length and width abnormalities occur, Butlin and Ensign or Magee or Kole or Egyedi-Obwegeser or Deplange or Pless or Austermann or Machtens designs and their modifi cations offer solution. For abnormally wide and thick tongue, dorsal resections are the only choice. In rare situations, when the length, width, and thickness are abnormal, Heggie’s Stellate anterior wedge or Davalbhakta and Lamberty design offer best possible solution. It has been already documented that there is no single ideal tongue resection procedure, rather the procedure needs to be customized considering the etiology, age, gender, existing dimension, and postoperative form/dimension desired. Such individual approach gives more pleasing results than the predetermined ones. In case of older individuals with mild to moderate enlargement, peripheral surgical trimming is ideal to prevent loss of taste as well as speech abnormalities. In case of extreme enlargement, AWR and CR can be chosen but with caution. The underlying disorder has to be corrected.                                                                                                                                                                    3. Overall, the article provides an insightful glimpse into the perioperative management of a pediatric patient with Beckwith-Wiedemann syndrome undergoing maxillofacial surgery. While the case presentation is detailed and informative, the article could benefit from providing more general context, simplifying technical language, addressing ethical considerations, and incorporating references to support the discussed practices.               3. Thanks for the valuable suggestion. From a technical point of view, we can consider the valutation that will be integrated with the characteristics of the various surgical techniques that can be used in this context: “In cases where length of the tongue is abnormal, resection designs of Reinwald or Gupta or Harris, Blair, Hendrick or Dingman and Grabb can be employed. In situations where abnormal width is a primary concern, Edgerton/Butlin and Handley/Pichler design or their modifi cation can be employed. For isolated thickness abnormalities, horizontal resections and key-hole debulking procedures are recommended. For situations where the length and width abnormalities occur, Butlin and Ensign or Magee or Kole or Egyedi-Obwegeser or Deplange or Pless or Austermann or Machtens designs and their modifi cations offer solution. For abnormally wide and thick tongue, dorsal resections are the only choice. In rare situations, when the length, width, and thickness are abnormal, Heggie’s Stellate anterior wedge or Davalbhakta and Lamberty design offer best possible solution. It has been already documented that there is no single ideal tongue resection procedure, rather the procedure needs to be customized considering the etiology, age, gender, existing dimension, and postoperative form/dimension desired. Such individual approach gives more pleasing results than the predetermined ones. In case of older individuals with mild to moderate enlargement, peripheral surgical trimming is ideal to prevent loss of taste as well as speech abnormalities.” We will also add that from a general and ethical point of view, the management of this type of procedure is particularly complex and involves the right tact and empathy for the relationship with the child but above all with her family unit, which certainly understands the importance, functional and health for the little patient but who probably does not imagine the possible risks associated with the operation and post-operative management. Furthermore, understanding the social context in which the child grows up is important, above all for the concern to follow her in rehabilitation, speech therapy, nutrition and social integration with her peers at school or in other fields.

Reviewer 4 Report

Dear Authors,

Thank You for manuscript correction.

Sincerely, Reviewer

Author Response

We thank the Reviewer for his/her positive comments.